# Spontaneous and Engineered Large Animal Models of Neurofibromatosis Type 1

**DOI:** 10.3390/ijms22041954

**Published:** 2021-02-16

**Authors:** Sara H. Osum, Adrienne L. Watson, David A. Largaespada

**Affiliations:** 1Masonic Cancer Center, Department of Pediatrics, Division of Hematology and Oncology, University of Minnesota, Minneapolis, MN 55455, USA; larga002@umn.edu; 2Recombinetics, Inc., Eagan, MN 55121, USA; adrienne@recombinetics.com

**Keywords:** neurofibromatosis type 1, large animal, spontaneous, genetically engineered, targeted therapy

## Abstract

Animal models are crucial to understanding human disease biology and developing new therapies. By far the most common animal used to investigate prevailing questions about human disease is the mouse. Mouse models are powerful tools for research as their small size, limited lifespan, and defined genetic background allow researchers to easily manipulate their genome and maintain large numbers of animals in general laboratory spaces. However, it is precisely these attributes that make them so different from humans and explains, in part, why these models do not accurately predict drug responses in human patients. This is particularly true of the neurofibromatoses (NFs), a group of genetic diseases that predispose individuals to tumors of the nervous system, the most common of which is Neurofibromatosis type 1 (NF1). Despite years of research, there are still many unanswered questions and few effective treatments for NF1. Genetically engineered mice have drastically improved our understanding of many aspects of NF1, but they do not exemplify the overall complexity of the disease and some findings do not translate well to humans due to differences in body size and physiology. Moreover, NF1 mouse models are heavily reliant on the Cre-Lox system, which does not accurately reflect the molecular mechanism of spontaneous loss of heterozygosity that accompanies human tumor development. Spontaneous and genetically engineered large animal models may provide a valuable supplement to rodent studies for NF1. Naturally occurring comparative models of disease are an attractive prospect because they occur on heterogeneous genetic backgrounds and are due to spontaneous rather than engineered mutations. The use of animals with naturally occurring disease has been effective for studying osteosarcoma, lymphoma, and diabetes. Spontaneous NF-like symptoms including neurofibromas and malignant peripheral nerve sheath tumors (MPNST) have been documented in several large animal species and share biological and clinical similarities with human NF1. These animals could provide additional insight into the complex biology of NF1 and potentially provide a platform for pre-clinical trials. Additionally, genetically engineered porcine models of NF1 have recently been developed and display a variety of clinical features similar to those seen in NF1 patients. Their large size and relatively long lifespan allow for longitudinal imaging studies and evaluation of innovative surgical techniques using human equipment. Greater genetic, anatomic, and physiologic similarities to humans enable the engineering of precise disease alleles found in human patients and make them ideal for preclinical pharmacokinetic and pharmacodynamic studies of small molecule, cellular, and gene therapies prior to clinical trials in patients. Comparative genomic studies between humans and animals with naturally occurring disease, as well as preclinical studies in large animal disease models, may help identify new targets for therapeutic intervention and expedite the translation of new therapies. In this review, we discuss new genetically engineered large animal models of NF1 and cases of spontaneous NF-like manifestations in large animals, with a special emphasis on how these comparative models could act as a crucial translational intermediary between specialized murine models and NF1 patients.

## 1. Introduction

The neurofibromatoses (NF) are a group of genetically distinct cancer predisposition syndromes that cause tumors of the central and peripheral nervous systems. There are three forms of NF: neurofibromatosis type 1 (NF1), neurofibromatosis type 2 (NF2), and schwannomatosis [1,2]. NF2 is caused by germline mutations of the *NF2* and is most commonly associated with bilateral vestibular schwannomas [2]. Schwannomatosis is clinically similar to NF2, but patients tend to develop schwannomas that spare the vestibular nerve [2]. While somatic *NF2* mutations are commonly found in these tumors, the genetic basis of schwannomatosis is more complex, and causative germline mutations have been found in both the *SMARC2B* and *LZRT1* gene [2,3]. Sporadic, NF2-associated, and schwannomatosis-associated schwannomas are indistinguishable aside from the molecular diagnosis [2]. NF1, the most common of the NFs, will be the main focus of this review.

Individuals with NF1 are born with heterozygous loss-of-function mutations in the *NF1* tumor suppressor gene, of which about half are familial and half are de novo mutations [2]. The *NF1* gene encodes the Ras GTPase activating protein (GAP) neurofibromin [4]. Loss of neurofibromin function causes hyperactive signaling through the MAPK and PI3K/mTOR pathways. NF1 syndrome is fully penetrant but the clinical manifestations of the disease are highly variable, even in patients with the same mutation [5]. Café au lait macules (CALMs) are the most common non-neoplastic manifestation, often noted at birth as areas of skin hyperpigmentation similar to a birthmark and characterized by loss of the remaining *NF1* wild type (WT) allele in melanocytes [6]. The most common tumor manifestations are benign peripheral nerve sheath tumors (PNSTs), which include benign cutaneous and plexiform neurofibromas as well as malignant PNST (MPNST) [7]. Similar to CALMs, biallelic inactivation, or loss of heterozygosity (LOH), of the *NF1* gene is required in Schwann cells (SCs) for the development of NF1-associated PNST [8,9]. Cutaneous neurofibromas are superficial growths that do not progress to malignancy but are a major source of morbidity for patients [9]. Plexiform neurofibromas, on the other hand, grow along deeper nerves, can grow quickly to impinge on major organs, and are often painful and disfiguring [10]. Between 10% and 30% of NF1-associated plexiform neurofibromas undergo transformation to malignant PNST (MPNST), a highly aggressive sarcoma with a poor 5-year survival rate of less than 40% [9,11]. NF1 patients are also at an increased risk for the development of other types of tumors, including breast cancer, optic pathway glioma (OPG), gastrointestinal stromal tumors, pheochromocytomas, and leukemias [1]. In addition to the high risk of tumor development, NF1 patients suffer from learning and behavioral difficulties, hypertension, skeletal abnormalities, and epilepsy [1,4,9].

Despite years of research to identify genes and pathways involved in disease progression and testing hundreds of drugs in various rodent models of NF1, very few therapies have made it to clinical trials, and NF1 continues to cause significant morbidity and a decreased life span in patients. The recent FDA approval of the mitogen-activated protein kinase kinase 1/2 (MEK1/2) inhibitor selumetinib for inoperable plexiform neurofibromas is one notable exception, although there are still improvements to be made with regard to adverse events and variability in response, and MEK inhibition may not be sufficient for other NF1-associated tumors like MPNST or juvenile myelomonocytic leukemia (JMML) [12,13,14]. Nonetheless, genetically engineered mouse models have significantly improved our understanding of NF1-associated tumor biology and have unique advantages for testing new prognostic and therapeutic tools [15]. Like all models, however, mouse models have constraints, including limited genetic heterogeneity, a short life span, and unreliable predictive validity [16]. Moreover, their small size precludes their use for imaging studies using human equipment and makes it difficult to perform pharmacokinetic (PK) and pharmacodynamic (PD) analyses, particularly in tissues. The predictive value of animal models is imprecise, and this is compounded by the complex, multisystemic nature of NF1. A combination of different models may therefore be required to better understand various aspects of NF1 and come closer to the clinical situation. In addition to mouse models, researchers should consider using alternative species to model and understand various aspects of NF1.

Naturally occurring large animal models provide an unprecedented opportunity to better understand genetic diseases due to their heterogeneous genetic backgrounds, long life span, large size, and more similar immune system to humans [17]. Indeed, many genetic diseases in domestic animals are currently being investigated as potential models for the human counterpart [18]. PNSTs, including schwannomas, neurofibromas, and MPNSTs have been reported in many large animal species, and NF-like syndromes are referred to in the literature, particularly in cattle and dogs [19,20]. The spontaneous occurrence and clinical similarity of these lesions to human neurofibromatosis could make them a valuable animal model to study genetic predisposition to cancer, but more research is needed to classify the genetic, molecular, and pathological basis of PNST in these animals and how they relate to NF1, NF2, and schwannomatosis in order to determine their effective use as comparative disease models.

In addition to naturally occurring models, advances in gene editing and reproductive technologies like CRISPR/Cas9 and somatic cell nuclear transfer (SCNT) have enabled researchers to develop precise models in more species than ever before, including domestic animals [21]. Several genetically engineered large animal models of human disease have been developed, particularly in pigs, and these models are becoming more widely used [22,23,24]. Pigs are already commonly used in translational research and are increasingly considered an optimal choice of nonrodent species for preclinical testing of new therapies [23]. They are significantly more similar than rodents to humans, particularly with respect to the nervous and integumentary systems [22]. Pigs have proven to be a useful animal model for human disease given their genetic, anatomic, and physiologic similarity to humans and have recently been shown to be an effective model of cancer [16,22,25,26,27,28]. We and others have recently developed a gene-edited pig carrying a germline heterozygous loss of function mutation in the *NF1* gene, and characterization of *NF1^+/−^* (NF1) minipigs strongly supports them as a comprehensive model for this disease [29,30]. In fact, this is the first animal model of NF1 to exhibit the spectrum of disease seen in human patients, including the development of both CALMs and spontaneous neurofibromas.

The last review discussing naturally occurring and genetically engineered animal models of neurofibromatosis was published by Riccardi, Womack, and Jacks in 1996, after the first transgenic mouse model of NF1 was developed and the first instance of a herd of dairy cattle with an apparently heritable form of neurofibromatosis was described [31]. Since that time, dozens of genetically engineered animal models of NF1 have been developed, and many more cases of spontaneous NF-like manifestations in animals have been reported. Our review will expand on this original review, discussing new potentially valuable large animal models of NF1 and highlighting the need for more studies using these models to test new potential therapies and address many questions that remain unanswered about this complex disease.

## 2. Spontaneous Models of Neurofibromatosis

Individual features of NF, particularly PNST, have been reported in many animal species. PNSTs are most commonly reported in cattle and dogs but have also been documented in other domestic species including cats, horses, goats, and pigs [32,33,34,35,36,37,38,39,40]. In humans, PNSTs are divided into subgroups including schwannomas, neurofibromas, MPNSTs, and a few other extremely rare tumors. Neurofibromas, commonly associated with NF1, are composed of a heterogeneous mixture of neoplastic Schwann cells and other non-neoplastic peripheral nerve components, including fibroblasts, perineurial cells, macrophages, and mast cells. Schwannomas are composed almost entirely of neoplastic Schwann cells and occur in patients with NF2 and schwannomatosis [41]. MPNSTs are aggressive sarcomas that can arise sporadically, but more commonly arise from plexiform neurofibromas in NF1 patients [11]. These tumors can be distinguished by histological and immunological criteria, which are critical for diagnosis, prognosis, and management of patients [41]. In the veterinary field, however, a consensus on the histological classification of PNSTs is lacking, and many case reports simply use the term benign and malignant PNST due to the similar clinical behavior between schwannomas and neurofibromas [32]. However, distinct similarities between subtypes of PNST in humans and animals have been shown, including tumor location, patient age, and histological criteria [32]. Other manifestations associated with NF are uncommon, and it has been difficult to prove that this is a heritable disorder in veterinary species. However, a genetic etiology of neurofibromatosis has been proposed in several animal species.

### 2.1. Cattle

PNSTs are one of the three most common neoplasms in cattle, accounting for up to 10% of bovine tumors found at slaughter [42]. The identification of multiple PNSTs is often an incidental finding at necropsy as clinical signs are typically only present in advanced cases due to organ or nerve compression [43]. The most common sites of involvement are the brachial plexus, heart, and intercostal nerves (Figure 1A) [44,45]. Many of these tumors closely resemble PNSTs associated with human neurofibromatosis grossly, histologically, and ultrastructurally (Figure 1A–D) [42,46]. Both schwannoma [44,47,48,49,50] and neurofibroma [50,51,52,53] are reported, but the distinction is not always clear in the literature, as lesions are often simply diagnosed as “neurofibromatosis” or PNST.

It has been proposed that cattle may have a hereditary predisposition to PNST, as in human NF. In 2014, Grossi et al. performed a statistical analysis on data from 567 cattle with PNSTs detected at necropsy [42]. Animals presented with multiple tumors, which were classified as schwannomas, hybrid schwannoma-neurofibromas, or neurofibromas based on immunohistochemical analysis [42,55]. Danish Holstein cattle were significantly more affected than other breeds examined, and a significantly higher number of affected offspring were from four Holstein sires, three of which were genetically related [42]. A genome-wide association study was then performed in 28 affected and 28 unaffected Holstein cattle, which identified a single nucleotide polymorphism at chromosome 27 associated with PNST. Cattle *NF1* and *NF2* map to chromosome 19 and 17, respectively. These results strongly suggest a hereditary predisposition to PNSTs in Holsteins. However, it is important to note that abattoir studies in cattle are limited because males are likely sent to slaughter before lesions become apparent [42]. Recently, Dammann et al. reported a high frequency of neurofibromas in the celiac ganglion of Holstein cattle [56]. Nervous tissues were collected from 403 Holstein Frisian cattle with the intention of finding potential risk factors for bovine spongiform encephalopathy (BSE), but the authors incidentally observed neurofibromas in 9.9% of animals upon histopathological analysis, specifically in the celiac ganglion. Interestingly, there was wide variation in neurofibroma incidence between cohorts ranging from 0 to 44.4%, providing further evidence of a genetic component to bovine PNST. However, immunohistochemistry of the lesions showed immunoreactivity of neurofibromin 1 and merlin proteins, making a genetic association with human NF less likely.

Unlike human NF1, cutaneous neurofibromas are rarely reported in cattle (Figure 1B–D). However, subpopulations of Holstein cattle develop cutaneous neurofibromas, and there is evidence that this may have a genetic etiology similar to NF1 in humans. For example, multiple cutaneous neurofibromas were reported in over 50 genetically related animals in a herd of Holstein cattle in Slovakia [57]. Similarly, cutaneous neurofibromas were observed in four cows from a Holstein dairy herd in Alabama with gross, histological, and ultrastructural features consistent with cutaneous neurofibromas found in NF1 patients [19]. These cows also presented with intraocular lesions similar to iris hamartomas found in human NF1, suggesting a similar multisystemic disorder. Three of the four affected cows were from the same sire, and linkage analysis revealed a polymorphism at the *NF1* locus in two cows and the sire, confirming inheritance of the paternal allele.

It is important to note that clusters of PNST cases in cattle from the same herd also support a viral etiology for some bovine PNSTs [58]. Virus-like particles have been identified in Schwann cells and fibroblasts from bovine PNSTs, but a viral origin has not been found despite attempts at virus isolation, immunohistochemistry (IHC), and animal inoculation [58,59]. Larger sequencing studies are warranted to determine the genetic and molecular underpinnings of PNST subtypes in cattle. These data will provide important comparative insights for the use of Holstein cattle as a preclinical model for human NF.

### 2.2. Dogs

In dogs, neurofibromas are typically associated with the skin, peripheral nerves, tongue, and intestine [32,60,61]. Although the generic diagnosis of benign PNST is often used, the existence of neurofibromas in dogs with similarities to humans has been described with regard to signalment, histology, immunohistochemistry, and electron microscopic findings [32]. Cutaneous neurofibromas are usually reported as singular masses, unlike those seen in humans [32,60,61]. However, multiple cutaneous neurofibromas were found in a seven-week-old black Labrador retriever who presented with an infiltrative intradural mass in the spinal cord consistent with rhabdomyosarcoma (RMS), as well as neuromelanocytosis (Figure 2A–C, cutaneous neurofibroma) [20]. While cutaneous neurofibromas are present around puberty in humans, children with NF1 are prone to the development of rhabdomyosarcomas [62]. The diagnosis of neuromelanocytosis may also be related to a defect in neurofibromin, as neurofibromin regulates melanocyte development, and children with NF1 show pigmentary abnormalities [1,63]. This combination of congenital abnormalities led the authors to suspect a germline mutation in *NF1*, but genetic testing was not performed due to financial constraints [20].

MPNST is more commonly reported than neurofibroma in dogs and is generally thought to be sporadic in origin [32]. However, there are several reports that arouse suspicion of an underlying predisposition syndrome. In one case, a ten-year-old Cocker spaniel presented with multiple masses in the lung and a cutaneous nodule on the left hindlimb (Figure 2D, lung mass) [61]. A diagnosis of MPNST with concomitant benign cutaneous PNST was made based on histopathological and immunohistochemical analysis (Figure 2E,F). The combination of benign cutaneous and malignant peripheral nerve sheath tumors is commonly seen in NF1 syndrome, although a distinction between schwannoma and neurofibroma was not determined in this case. In another case, a three-month-old golden retriever dog presenting with right hind limb enlargement was diagnosed with a benign PNST based on histology (Figure 2G) [64]. Rapid progressive clinical signs over seven months led to amputation of the limb and final diagnosis of MPNST of the sciatic nerve, suggesting malignant transformation (Figure 2H) [64]. The dog also showed evidence of skeletal abnormalities including tibial bowing, which is seen in NF1 patients, although this could have been a consequence of the tumor [1,64]. The age of the patient, tumor location, and evidence of progression from benign neurofibroma to MPNST are analogous to NF1-associated MPNST, which most commonly arise from benign congenital plexiform neurofibromas associated with deep nerves. However, genetic testing was not performed, and the lesion was not distinguished from schwannoma. Further examples of benign neurofibroma to MPNST transformation were shown in an immunohistochemical study of canine MPNST and PNST where several benign PNST showed histologic patterns similar to human atypical neurofibroma and evidence of malignant transformation [60]. Prediction of malignant transformation from plexiform neurofibroma in NF1 patients is currently a major challenge for clinicians [65]. Comparative genetic studies in canine PNST could increase our understanding of malignant transformation in human neurofibromas and thereby improve early detection and management of these aggressive tumors.

Finally, a comparative preclinical trial of intratumorally injected attenuated *Clostridium novyi* spores was performed in 10 dogs with soft tissue sarcomas, seven of which were PNSTs [66]. Exome sequencing of tumor and matched normal DNA showed somatic mutations in several genes commonly mutated in human PNSTs, including *NF1*. It is unclear whether there was a concurrent germline mutation in *NF1*, but the numbers of genetic alterations and spectrum of mutations in canine PNST were similar to those of humans, further demonstrating the complementarity between human and canine PNST. In fact, the promising results from this preclinical study led to a phase I investigational study, in which a patient received the same therapy and experienced a similarly robust anti-tumor response (NCT01924689). The results of this study suggest that preclinical studies in spontaneous PNSTs from dogs can provide a complement to other preclinical animal models and help guide studies in humans.

Preclinical studies of PNST in dogs are of interest due to their heterogeneous genetic backgrounds, intact immune system, and spontaneous rather than engineered origins. However, larger studies are needed to fill the gaps in our knowledge of canine PNST. Immunohistochemical evaluation of canine PNST is becoming more widespread, but there is still no consensus on diagnostic criteria, and the true incidence of neurofibroma in dogs and other animals is unknown [32,60]. Genetic testing for underlying genetic mutations predisposing to neurofibroma development would be valuable, particularly in young dogs that present with PNSTs in combination with other abnormalities typically associated with NF1, as specific dog breeds may have a significant and reproducible predisposition for developing certain features of NF1. Further studies are also warranted to determine whether canine neurofibromas have the potential to undergo malignant transformation. Larger-scale comparative genomic studies between human, canine, and rodent models of NF1 would be of value to help identify commonly affected and targetable pathways that may serve as drug targets or potential biomarkers for NF1 patients.

### 2.3. Other Spontaneous Animal Models

While the topic of this review is focused on large animal models, a brief review of interesting cases is warranted, particularly the bicolor damselfish. In 1983, a subpopulation of damselfish in southern Florida was reported with pigmentary abnormalities and multiple PNSTs [67]. This was the first report of animals with a naturally occurring and persistent NF1-like syndrome. As with cattle neurofibromatosis, both a viral and a genetic etiology have been proposed, but not shown definitively [68,69]. Interestingly, multiple benign cutaneous PNST were reported in two related bearded dragons, raising the question of whether a heritable syndrome similar to human neurofibromatosis may be present in lower vertebrates as well. Porcine PNSTs have been documented rarely but include cutaneous schwannoma, pigmented cutaneous neurofibroma (Figure 3A–C), and MPNST (Figure 3D–F) [36,37,38].

## 3. Genetically Engineered Models of NF1

### 3.1. NF1 Genetically Engineered Rodent Models (NF1-GERMs)

Recent decades have seen revolutionary advances in genetic engineering technologies for animal modeling of human disease. The first genetically engineered mouse models (GEMMs) for NF1 were described in 1994 and carried a germline inactivating mutation resulting in a non-functional *Nf1* allele [70,71]. Unfortunately, *Nf1^+/−^* mice did not develop PNSTs or other classical symptoms of NF1 syndrome described in humans, and homozygous mutants died in utero due to cardiac defects [70,71]. Nevertheless, these initial mouse models paved the way for more sophisticated modeling systems, including chimeric *Nf1^−/−^* mice, some of which developed plexiform neurofibroma-like lesions, and *Nf1^+/−^* and *Trp53^+/−^* (NPcis) mice, which rapidly developed MPNST-like tumors [72,73]. Still, these mouse models did not develop more common clinical manifestations like café au lait macules, cutaneous neurofibromas, or Lisch nodules. Moreover, MPNSTs from NPcis mice did not develop from pre-existing plexiform neurofibromas, which commonly involves progressive loss of *CDKN2A* and *SUZ12* or *EED* in NF1 patients [74,75]. To better model this malignant transformation, mice with germline-inactivating mutations in *Suz12* and *Nf1* were generated and showed accelerated formation of neurofibromas and MPNST-like tumors [76]. The development of other tumors and the stochastic nature of tumor formation in this model suggest that the location and timing of these mutations are critical for modeling the human counterpart. The advent of the Cre-lox system has enabled researchers to generate tissue-specific and inducible mutations, allowing for control over the location and timing of gene expression. Many groups took advantage of this system to generate mice with specific manifestations of NF1, using various Cre drivers to generate mice with neurofibromas, OPG, JMML, and other NF1 features (Table 1). This system has proved essential to harnessing the temporal and tissue-restrictive nature of NF1-associated tumors to achieve sophisticated rodent models that develop solitary features of NF1. Aside from technical problems associated with the Cre-lox system, the main limitation of these tumor models is that they do not mimic the spontaneous LOH associated with human tumor development.

NF1-GEMMs have been the cornerstone for the development and testing of new therapies for NF1-associated neoplasia. For example, a preclinical trial of the MEK inhibitor selumetinib in the Dhh-Cre;Nf1^flox/flox^ GEMM showed pharmacokinetic results similar to those in humans with subsequent decreases in baseline neurofibroma tumor volume in 67% of animals by volumetric magnetic resonance imaging (MRI) [90]. This proved to be predictive of the human response, and selumetinib is now the first and only FDA-approved therapy for NF1 [12]. Conversely, the tyrosine kinase inhibitor sorafenib demonstrated significant decreases in tumor volume in Dhh-Cre;Nf1^flox/flox^ GEMM neurofibromas, but PK parameters did not align with human patients, and children with NF1-associated plexiform neurofibromas did not tolerate sorafenib at doses significantly lower than those of non-NF1 children with other advanced cancers [91,92]. Similarly, several promising therapeutic targets have been identified using the NPcis MPNST mouse model, including epidermal growth factor receptory (EGFR) and mechanistic target of rapamycin (mTOR), but none have demonstrated activity in clinical trials for MPNST [14,93,94]. The variable predictive value of preclinical NF1 mouse models is due in part to the limited genetic diversity of these systems, which do not fully recapitulate the genetic complexity of the human tumors. Naturally occurring models of MPNST and neurofibroma could shed light on these differences and provide a complementary preclinical model for therapeutic drug development in neurofibromatosis.

Learning and memory deficits are common in patients with NF1 [95]. While the *Nf1*^+/−^ mouse model does not develop classical symptoms of NF1, a subset of mice develop learning, memory, and attention deficits similar to human NF1 [96,97]. It was later shown that learning deficits in *Nf1*^+/−^ mice may be due to increased Ras/MAPK activity leading to increased GABA-mediated inhibition, which could be reversed by genetic or pharmacological inhibition of these pathways [98]. Indeed, lovastatin decreased MAPK activity in *Nf1*^+/−^ mice and reversed their learning and attention deficits [96]. These results led to clinical trials of statins, but they did not improve learning in children with NF1 [99]. Rodent models have been shown to be valuable tools for studying behavioral and cognitive deficits in a variety of clinical disorders, but many of these impairments involve higher cognitive functions that are difficult to study in rodents; therefore, their translational potential for NF1-targeted behavioral therapies is lacking [100].

NF1 patients have a five-fold increased risk of developing breast cancer, and individuals with sporadic breast cancer commonly have mutations in *NF1* [9]. Dischinger et al. utilized the CRISPR-Cas system to develop a rat model of NF1 [101]. Targeting the GAP-related domain of *Nf1* in Sprague-Dawley rats, they generated rats with either in-frame deletions or nonsense mutations [101]. *Nf1* rats developed aggressive, estrogen-dependent mammary adenocarcinomas within 6–8 weeks of age [101]. This new model enables researchers to investigate the interplay between NF1 and estrogen receptor (ER) signaling in sporadic and NF1-associated breast cancer and underscores the importance of alternative animal models.

While GERMs have provided valuable insights into the cellular and molecular determinants that underlie rodent NF1 tumor development and maintenance, they do not mimic the disease course or represent the complex clinical background of NF1 patients. Differences in physiology, anatomy, size, and lifespan between mice and humans make them well-suited for certain preclinical studies and not for others. For example, many manifestations of NF1 develop over longer time frames which may not be feasible to study in mice due to their short life span. However, short-lived mice are able to recapitulate the human condition of a rare cell undergoing LOH using Cre-Lox technology, which allows for biallelic deletion of *Nf1* in specific subsets of cells, while other cell types remain heterozygous or wild type for *NF1.* The unique advantages and disadvantages of rodent models reinforce the need for complementary preclinical studies in different animal species with more similar body size and organ systems, particularly studies requiring intact immune systems, human-sized equipment, or serial blood and tissue sampling.

### 3.2. NF1 Genetically Engineered Swine Models

Recently, powerful genetic engineering methods and advanced reproductive technologies have expanded to more physiologically relevant species, particularly swine, enabling researchers to develop more representative clinical models of disease. Swine have already proved to be useful in a wide range of preclinical research. Given their similarity to humans in terms of body size, they are commonly used to study human-sized imaging equipment and medical devices, and their relatively long lifespan allows for longitudinal studies. They are also commonly used for toxicology of pharmaceuticals prior to clinical trials. Spontaneous cancer in pigs is rare, although the Libechev and Sinclair minipigs are predisposed to developing melanoma and have been used to study human melanoma [25]. Production of genetically modified pigs is more labor- and time-intensive than of mice, but the potential benefits for preclinical research are substantial. The use of genetically engineered pigs has proven valuable for several genetic diseases, including diabetes and cystic fibrosis, and the development of porcine cancer models has also increased dramatically (Figure 4) [22,25,101]. Swine are an attractive model for NF1 syndrome in particular, due to their similarities with regard to their nervous, integumentary, and cardiovascular systems [23,102].

We and others have recently developed genetically engineered swine models of NF1 syndrome [29,30]. In an effort to produce an animal model that accurately represents the human condition, we sought to engineer minipigs with a specific patient mutation. Since there are no hotspot mutations in NF1, we chose *NF1^R1947X^*, a nonsense mutation that has been found in a relatively large subset of NF1 patients and is associated with a wide spectrum of disease including neurofibromas, optic gliomas, and skeletal abnormalities [29,104]. We designed transcription activator-like effector nucleases (TALENs) to target a region in exon 41 of the swine *NF1* gene, which shares 100% amino acid homology with human exon 39, and then used a homology-directed repair (HDR) construct to introduce the *NF1^R1947X^* point mutation into embryonic fibroblasts from Ossabaw minipigs. Heterozygous clones were isolated, and clonal *NF1^+/R1947X^* offspring were generated by SCNT [105]. *NF1^+/R1947X^* minipigs were viable and exhibited germline transmission with Mendelian frequency. We observed 100% penetrance of café au lait macules (CALMs), a phenotype that has not been demonstrated in other animal models. Similar to what is seen in human patients, these lesions were present at birth, increased in size and number over time, and did not progress to malignancy. A subset of *NF1^+/R1947X^* minipigs also developed cutaneous masses that grossly and histologically resemble cutaneous neurofibromas found in a majority of NF1 patients (Figure 5). Neurofibroma-derived Schwann cells and CALM-derived melanocytes showed LOH at the *NF1* locus with consequent loss of neurofibromin protein expression and Ras hyperactivation [6,106]. This spontaneous LOH in Schwann cells and melanocytes has never been shown in mouse models. The ability to biopsy tissues and culture primary cells for in vitro studies is a valuable tool and a key feature of the NF1 minipig model. In addition to primary melanocytes and Schwann cells, we have successfully established primary cultures of mast cells from bone marrow, fibroblasts from normal nerve and neurofibromas, and PBMCs using pig-specific or species cross-reactive growth factors (unpublished data). These cells can be used for downstream in vitro analyses to evaluate cell-specific reactions to targeted therapies as well as to develop cell–cell interaction models in two and three dimensions. Electron microscopic analysis of optic nerves from *NF1^+/R1947X^* minipigs showed myelin decompaction, which was described in enlarged brain white matter tracts from mice with astrocyte-specific loss of *Nf1* and has been associated with behavioral deficits and vision changes in NF1 patients [107,108]. MRI and computed tomography (CT) also revealed evidence of OPG, which are commonly found in pediatric NF1 patients [5]. Tibial narrowing was also noted in a subset of *NF1^+/R1947X^* minipigs.

Healthy minipigs are increasingly being used to test the safety and toxicity of investigational drugs prior to clinical trials (Figure 4), so the concept of expanding their use to preclinical PK and PD in a disease model is appealing. The minipig is an optimal species for PK and PD because its large size allows for collecting large amounts of blood and tissue for downstream analyses, and their metabolic and genetic similarities to humans enable researchers to readily translate drug metabolism and target engagement of molecular therapies to inform clinical trials. To evaluate the behavior of NF1-targeted therapies in NF1 minipigs, we first tested a commercially available drug, gabapentin, which is used to treat neuropathic pain in NF1 patients. Steady-state PK of gabapentin in *NF1^+/−^* and WT littermate controls showed peak plasma concentrations that are efficacious in humans, and a similar PK profile (unpublished) [90,109]. We next evaluated the PK and PD of several MEK inhibitors that have shown promise in patients with NF1-associated nervous system tumors. Single-dose PK of PD0325901 showed a similar PK profile to NF1 patients in clinical trials and exhibited a robust PD effect in PBMCs isolated from treated minipigs [29]. Selumetinib, the first FDA-approved therapy for NF1, was also evaluated in *NF1^+/R1947X^* and WT littermates, and plasma PK and PD profiles of selumetinib aligned with those of NF1 patients [90,110,111,112,113]. We also had the unique opportunity to evaluate the pharmacology of selumetinib in tissues clinically relevant to NF1, including cerebral cortex, optic nerve, sciatic nerve, and skin. Selumetinib was detectable in all tissues within two hours of oral administration, and significant inhibition of p-ERK was observed in skin and sciatic nerve from all minipigs [110]. Interestingly, basal p-ERK levels were significantly higher in *NF1^+/R1947X^* minipig optic nerve compared to WT, which were efficiently reduced to WT levels with selumetinib [110].

While molecularly targeted therapies are showing promise in clinical trials, surgery is still the mainstay of treatment for amenable NF1-associated benign and malignant tumors [1]. However, complete surgical excision can be difficult or impossible due to the size, number, or close association of tumors with surrounding tissues, and tumor re-growth or metastasis is common [9]. Swine are widely considered the default model for both surgical training and research, as their size and anatomy enable the use of surgical approaches intended for use in human patients, thus facilitating clinical translation [102]. The NF1 minipig could therefore be a valuable resource for testing new surgical interventions for NF1 patients. For example, surgical approaches for many of the cutaneous manifestations of NF1 carry a high risk of scarring and delayed wound healing [9]. Given their similar skin anatomy, swine are commonly used to study wound healing and reconstructive surgery [102]. The NF1 minipig develops cutaneous neurofibromas and café au lait macules and could therefore be used to evaluate new strategies to reduce scarring and improve wound healing in NF1 patients. For more invasive tumors associated with vital nerves, it is likely that a multimodal approach including both molecular and surgical approaches will be critical to ensure complete regression and prevent regrowth. The minipig could be a useful tool for evaluating the effectiveness of molecularly targeted therapies as adjunctive therapy before or after surgical intervention in these patients.

Prophylactic treatment to reduce or remove the risk of tumor development has recently been considered. Chemoprevention has proven effective for some cancers, as exemplified by the HPV vaccine in preventing cervical cancer [114]. However, the effectiveness of chemoprevention is notoriously difficult to document, mostly due to the long latency of cancer and the requirement for therapies to be well tolerated and safe for long durations. Chemoprevention in NF1 is an attractive concept but has not been thoroughly studied. Recently, Staedke et al. showed that mebendazole, alone or in combination with cyclo-oxygenase-2 inhibitors, could prevent MPNST development in the NPcis mouse model of NF1 [115]. The NPcis mouse develops MPNST rapidly, whereas human patients can take years to develop tumors, making this therapy difficult to evaluate in both mice and humans. The NF1 minipig develops tumors within 4–8 months of age, making it an ideal preclinical model for longitudinal safety and efficacy studies over a considerable amount of time to ensure that a chemopreventative is safe for long term consumption.

In addition to preclinical pharmacology studies, the NF1 minipig is a valuable comparative model for studying behavioral and cognitive manifestations of NF1 syndrome in humans. White et al. developed a genetically engineered minipig model of NF1 utilizing recombinant adeno-associated virus–mediated gene targeting and SCNT to generate *NF1^+/ex42del^* minipigs with a complete deletion of the endogenous *NF1* exon 42 [30]. *NF1^+/ex42del^* minipigs exhibited manifestations of NF1 syndrome similar to *NF1^+/R1947X^* and humans, including CALMs, neurofibromas, tibial narrowing, axillary/inguinal freckling, and shortened stature [116]. T2 hyperintensities were also noted in *NF1^+/ex42del^* minipigs, which have been associated with cognitive impairment in individuals with NF1 [30]. Correspondingly, *NF1^+/ex42del^* minipigs also showed learning and memory defects seen in NF1 patients and mouse models, including initial learning delays in a *T*-maze test [30]. Overall observations of hyperactivity and anxiety were also noted in *NF1^+/ex42del^* minipigs, which may correlate with symptoms of NF1 as there is a high prevalence of attention-deficit/hyperactivity disorder (ADHD) and autistic traits reported in children with NF1 [30,95].

In addition to the more commonly described symptoms of NF1 syndrome, up to 70% of NF1 patients report chronic pain as a key symptom of the disease, mainly due to complications of NF1 including tumor manifestations and skeletal deformations [117]. Chronic pain can substantially affect quality of life, yet this is a highly understudied phenomenon in the NF1 population. Current therapies for NF1-derived pain include generic pain medications and removal of the inciting tumor. NF1 clinical trials have begun to incorporate pain reduction in patient-reported outcomes, but measuring pain in patients remains a challenge [117]. The NF1 minipig may be a useful model for studying NF1-derived pain, as initial studies in *NF1^+/ex42del^* minipigs showed evidence of decreased pain thresholds, including allodynia and hyperalgesia, in female *NF1^+/ex42del^* pigs [118]. Additionally, tumor-bearing males showed impaired sleep quality and increased resting, both of which have been reported in NF1 patients with chronic pain [118]. Little is known about the molecular mechanisms of NF1-associated pain, but studies in mice suggest that neurofibromin may play a key role in the excitability of nociceptive sensory neurons. Sensory neurons from *Nf1^+/−^* mice exhibit increased excitability, increased sodium peak current densities, and dysregulated calcium signaling [119,120,121]. These factors may explain the painful conditions experienced by people with NF1. Interestingly, dysregulation of *N*-type voltage-gated calcium (CaV2.2) and sodium channels was also reported in *NF1^+/ex42del^* minipigs, and increased activity of these calcium channels could be suppressed by pharmacological antagonism of collapsin response mediator protein 2 (CRMP2) phosphorylation, which is known to interact with neurofibromin [118]. The comparative similarities between NF1 mouse and pig models in nociceptive dysregulation are strongly suggestive of a role for neurofibromin in chronic pain. Since the NF1 minipig develops co-morbidities associated with NF1 including tumors and sleep deficits not seen in mouse models, this model could provide valuable insights into the molecular mechanisms of nociception in NF1 patients with implications for future therapeutic interventions.

Finally, there is a pressing need for new imaging methods that can be performed more easily and be incorporated into routine clinical practice. Whole-body MRI is a critical diagnostic for NF1 patients and can identify and localize plexiform neurofibromas, gliomas, and MPNSTs [122]. Medical imaging is also crucial for the management of tumors in NF1 patients and for assessing treatment response in clinical trials [123]. However, management of patients at risk for MPNST is challenging, and standardized, cost-efficient imaging methods for routine screening are needed [91]. OPG presents a similar challenge, as tumors often remain asymptomatic but their natural history is unpredictable [9]. Currently, children with OPG are evaluated annually with a barrage of vision tests as MRI screening does not affect outcome [9]. These tests require the child’s cooperation, which can be challenging in pre-verbal children with NF1-associated co-morbid cognitive and behavioral deficits. Optical coherence tomography (OCT) is a relatively new non-invasive imaging technology that has shown promise in the diagnosis and evaluation of OPG-associated pathology [124]. However, this technique requires further validation with regard to defining clinical progression and determining which children require treatment. Developing standardized imaging protocols and optimizing the use of alternative screening techniques like OCT in a swine model may help clinicians monitor disease progression in patients with NF1.

The NF1 minipig models provide a unique platform to investigate unanswered questions regarding the relationship between T2 hyperintensities and behavioral abnormalities in children in addition to developing new imaging strategies for early detection of NF1 tumors and longitudinal assessment of tumor dynamics [29,30,116]. Additionally, long-term prospective natural history studies are needed to better characterize tumor growth and functional outcomes, as current studies are limited to relatively short observation periods [123]. Advanced imaging technologies can also be used to study learning disabilities, attention deficits, and social and behavioral problems that decrease the quality of life of NF1 patients [125]. The gross anatomical features and development of the minipig brain are similar to those of humans, and their large size allows for the use of neuroimaging instruments designed for humans to be used with adult pigs and piglets [126,127]. Consequently, the NF1 minipig provides the ideal platform for developing new imaging strategies for the detection and longitudinal evaluation of NF1 tumor dynamics. Moreover, cognitive dysfunction in NF1 patients has been linked to the presence of T2 hyperintensities in the brain, which could be further explored in the minipig [128].

These results suggest that NF1 minipigs and the cell lines generated from their tissues will be useful in answering prevailing questions in the field of NF1 and may facilitate the development of new imaging and surgical modalities in addition to new targeted therapies for NF1-associated nervous system tumors in humans. There are many challenges remaining to be solved in NF1 for which the pig could be used. As a preclinical pharmacology model, the pig can be used to test new MEK inhibitors alone, in combination, and as adjunctive therapy prior to surgery, as well as long-term safety, efficacy, and tolerability of different classes of drugs such as prophylactic treatments and cell and gene therapy. They can also be used during and after clinical trials to optimize dosing and scheduling for NF1 patients. Tissues and primary cells can be isolated from the pigs to determine drug tissue penetrance and target cell-specific vulnerability. These cells and tissues can also help answer basic biological questions about NF1 syndrome, like how different cells in the tumor microenvironment interact to promote tumor growth and how we can target these interactions to halt or regress tumors. Pigs with tumors can also be used to test new imaging and surgical techniques prior to use in the clinic. The pig is uniquely poised to complement genetically engineered mouse models and spontaneous models of NF, but time and resource limitations often preclude their widespread use, thus impeding valuable preclinical research. It is likely that specialty companies and contract research organizations will be required to allow for scientists to use minipigs effectively to study various aspects and stages of disease over time.

## 4. Conclusions

Animal models are essential to our understanding of NF and for translating basic research findings into use in the clinic. Since the last publication on naturally occurring and genetically engineered models of neurofibromatosis, we have developed extremely sophisticated and powerful tools to better understand human disease using animal models. From precision gene-editing technologies, advanced reproductive technologies such as cloning, and next-generation sequencing to advanced medical imaging and surgical techniques, we have made great strides in better diagnosing and managing NF1 patients. However, NF is a complex and heterogeneous group of diseases, and there are still many unanswered questions in the field. Advanced technologies are increasingly being utilized in clinical trials and small animal models, and researchers could take advantage of these tools to develop improved spontaneous and genetically engineered large animal models of NF. Additionally, classification and terminology recommendations from human and veterinary neuropathologists will be essential for the validation of these models and the widespread acceptance of their use in comparative preclinical research. An integrated and collaborative preclinical approach using naturally occurring and genetically engineered small and large animal models will ultimately enable the discovery and development of more predictive therapies and diagnostic techniques and increase the success rate of drugs that enter clinical trials.

## Figures and Tables

**Figure 1 ijms-22-01954-f001:**
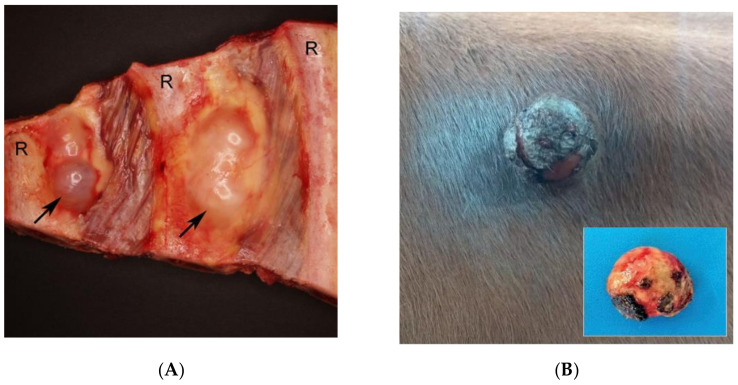
Spontaneous features of neurofibromatosis (NF) in cows. Bovine peripheral nerve sheath tumor (PNST) (**A**) and cutaneous neurofibroma (**B**–**D**). (**A**) PNST. Two tumors (arrows) located in relation to intercostal nerves (not visible). Specimen displaying the internal surface of the thorax. R: rib [42]. (**B**) Cutaneous neurofibroma. Note the firm exophytic nodule with marked ulceration in the paralumbar fossa. Inset: Nodule after complete surgical excision [54]. (**C)** Cutaneous neurofibroma. Note the mesenchymal cell proliferation forming plexiform beams. Hematoxylin-Eosin. ×200 [54]. (**D**) Cutaneous neurofibroma. Positive immunoblotting for S-100, counterstained with Harris Hematoxylin. ×400 [54].

**Figure 2 ijms-22-01954-f002:**
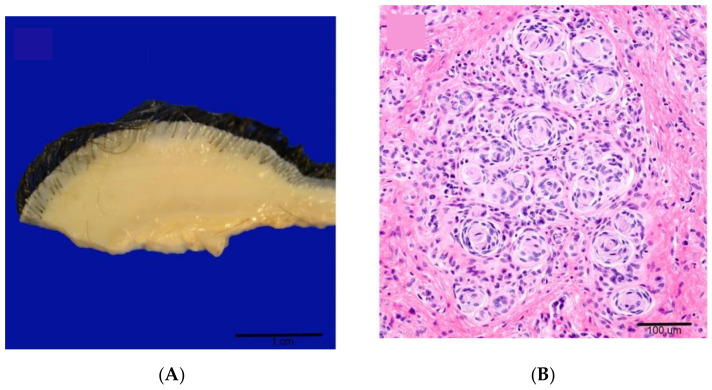
Spontaneous features of NF in dogs. Cutaneous neurofibroma (**A**–**C**), malignant peripheral nerve sheath tumors (MPNST) (**D**–**F**), and malignant transformation to MPNST (G-H). (**A**) Gross appearance of a neurofibroma effacing the dermis and subcutis of haired skin [20]. (**B**). Tactile-like structures characterized by parallel stacks or whorls 3–7 cells thick surrounded by a perineurial cell capsule (pseudomeissnerian corpuscles) in cutaneous neurofibroma. H&E [20]. (**C**) S100 immunohistochemistry (IHC) of cutaneous neurofibroma reveals positive staining of the pseudomeissnerian corpuscles (star) but lack of stain uptake in the perineurial cells (arrow) [20]. (**D**) Pulmonary MPNST in Cocker Spaniel dog with concomitant cNF. White multilobular mass in the caudal right lung [61]. (**E**). Section from the pulmonary mass showing dense proliferation of neoplastic cells arranged in interwoven bundles and concentric whorls (Antoni A pattern). H&E [61]. (**F**). Section from the pulmonary mass showing strong expression of S100 IHC [61]. (**G**) Golden Retriever dog with right hind limb enlargement. The sciatic nerve is infiltrated by a hypocellular well-differentiated spindle cell neoplasm diagnosed as benign PNST. Bar = 100μm [64]. (**H**) The large soft tissue mass surrounding the right ischium is composed of sheets of malignant spindle cells. Bar = 100 μm [64].

**Figure 3 ijms-22-01954-f003:**
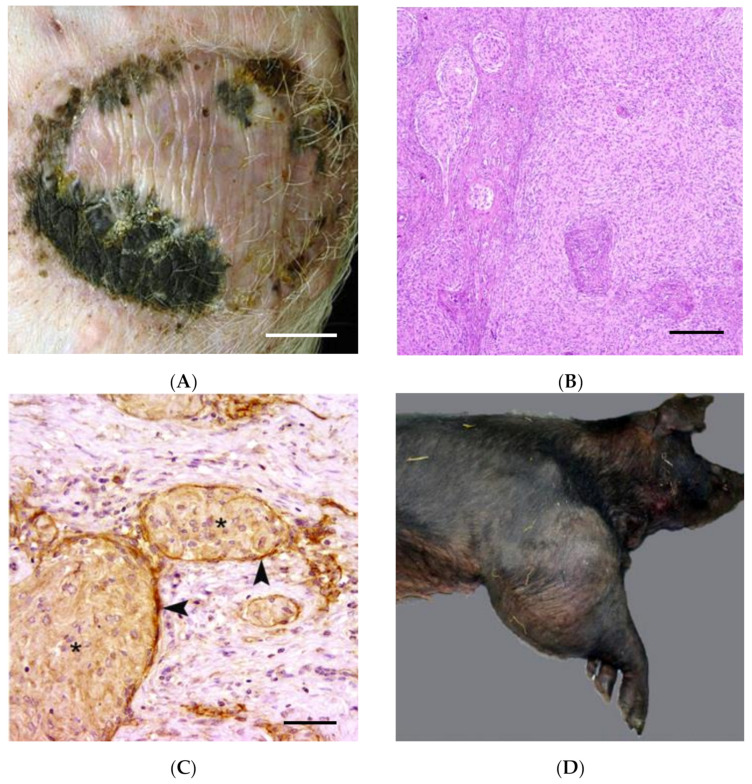
Spontaneous features of NF in pigs. Cutaneous pigmented neurofibroma (**A**–**C**) and MPNST (**D**–**F**). (**A**) Pigmented mass in the skin of the ventral abdomen. Bar, 3 cm [38]. (**B**) Cutaneous pigmented neurofibroma. The non-pigmented spindle cell population forms fascicles, bundles, and variably sized nodules. Bar, 100μm [38]. (**C**) Cutaneous pigmented neurofibroma Perineurial-like cells (arrowheads) and neoplastic Schwann cells (asterisks) express nerve growth factor receptor (NGFR P75). IHC. Bar, 50μm [38]. (**D**) MPNST. A firm mass expands the right humerus, elbow joint, and proximal radius and ulna [40]. (**E**) MPNST. The neoplasm is composed of closely packed, interweaving short streams and bundles of spindle cells in a fine fibrovascular stroma. H&E [40]. (**F**) MPNST. Closer view of the neoplasm with interweaving streams and bundles of pleomorphic spindle cells. H&E [40].

**Figure 4 ijms-22-01954-f004:**
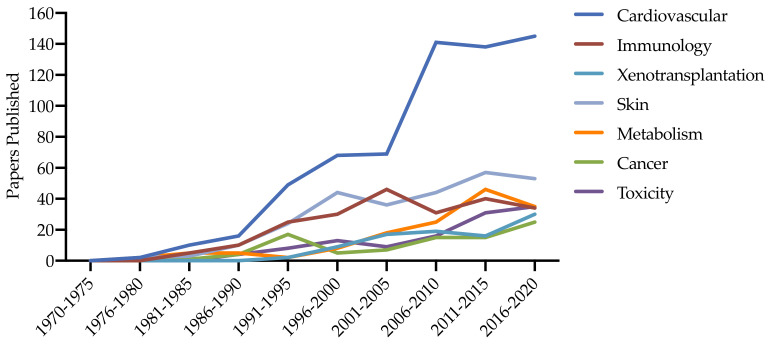
The use of minipig biomedical models over time based on articles indexed by PubMed from 1970 to 2020 [103].

**Figure 5 ijms-22-01954-f005:**
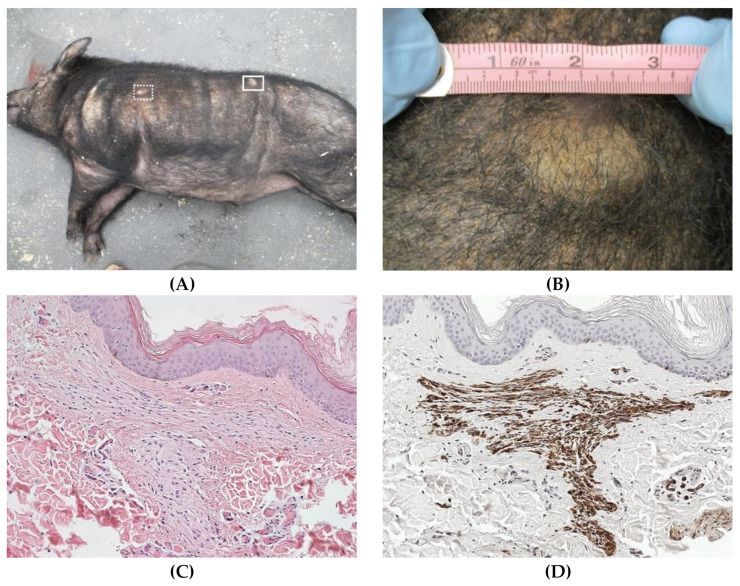
Genetically engineered NF1 minipigs develop neurofibromas [29]. (**A**) An example of an NF1 minipig harboring two dermal masses on its left side. The mass denoted by the white box is enlarged in B. (**B**) mass on the flank measured 4.2 cm in diameter. (**C**) H&E staining of a representative mass showing regions of hypercellularity. (**D**) Hypercellular regions stain positive for S100β, a marker of Schwann cells.

**Table 1 ijms-22-01954-t001:** GEMM of NF1-associated clinical manifestations using the Cre-Lox system.

Genotype	Cell Type	Phenotype	Reference
GFAP-Cre; Nf1^flox/mut^	Astrocytes	Optic pathway glioma	[77]
P0A-Cre;Nf1^flox/−^	Nonmyelinating SCs	Plexiform neurofibroma	[78]
Plp-CreER;Nf1^flox/flox^	Schwann cell precursors (SCPs)	Plexiform neurofibroma	[79]
Krox20-Cre;Nf1^flox/−^	SCPs	Plexiform neurofibroma	[80]
Dhh-Cre;Nf1^flox/flox^	SCPs	Plexiform/cutaneous neurofibroma	[81]
Dhh-Cre;Nf1^flox/flox^; Ink4a/Arf^+/−^; Ink4a/Arf^−/−^; Ink4a/Arf^flox/flox^	SCPs	atypical neurofibroma, MPNST	[82]
CMV-CreER T2;Rosa26;Nf1^flox/−^	Skin-derived precursors (SKPs)	Cutaneous neurofibroma	[83]
Hoxb7-Cre;Nf1^flox/flox^;LacZ	SKPs	Cutaneous/plexiform neurofibroma	[84]
Prss56-Cre;Rosa26-tdTom;Nf1flox/flox	Boundary cap cells	Cutaneous/plexiform neurofibroma	[85]
Postn-Cre;Nf1^flox/flox^;Arf ^flox/+^	SCPs	Plexiform neurofibroma, atypical neurofibromatous neoplasms of uncertain biologic potential (ANNUBP), MPNST	[86]
Mx1-Cre; Nf1^flox/flox^	Bone marrow	JMML	[87]
CAD5-Cre^ERT2^/Rosa26-LSL-td-Tomato; Nf1^flox/flox^	Vascular endothelium	Altered proliferation and morphogenesis	[88]
Col2α1-Cre; Nf1^flox/flox^	Chondrocytes	Skeletal dysplasia	[89]

## Data Availability

The data presented in this study are available on request from the corresponding author.

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
