# Peer review of "Spontaneous and Engineered Large Animal Models of Neurofibromatosis Type 1"

_ijms, 2021, doi:10.3390/ijms22041954_

Round 1

Reviewer 1 Report

The authors present an interesting narrative review about the value of large animal models of neurofibromatosis. The review is intriguing and well-written, and potentially relevant for the readers; nevertherless, I have some issues:

The first one is related to the introduction, mainly from lines 49 to 107: I think that this section is really too long, focused on human clinical and molecular description of RASopathies in human (not strictly the article focus); therefore, I suggest, if possible, to summarize it

Fig 1 is quite confusing, also in part for the low quality of the pictures in the panel (whereas i understand that are reproducted from an old article). In example, D: in black and whit is not easy to fully appreciate the cutaneus NF (if color figure is not available, arrows or other sign could be helpful). Is this figure striclty necessary?

Line 604: MRI and CT are also a critical diagnostic for NF1 patients and can 603 identify and localize plexiform neurofibromas, gliomas, and MPNST [125]; however, ref 125 (Khanna et al, Assessment of nociception and related quality-of-life measures in a porcine model of neurofibromatosis 974 type 1) does not appear fitting with the diagnostic evaluation in clinical practice. Please correct

Lines 604-606: While imaging is crucial for early detection of MPNSTs, routine screening is currently not recommended for asymptomatic patients. I only partially agree: a well-defined follow-up is part of managment of patients with RASopathies, mainly NF. Please clarify

Disclosure section and Conflicts of Interest section are missing: these section are mandatory, and authors should clarify the exact role of the company involved in the article (Recombinetics, Inc., affiliation of the second author) in the manuscript 's drawing.

Author Contributions section is missing too.

Reviewer 2 Report

This study (manuscript ijms-1087089) by Osum et al. is a review article describing the different spontaneous and genetically engineered animal model of neurofibromatosis type 1 (NF1).

The manuscript is well written and it is an interesting and original review.

The general theme of the review is little covered by literature, whereas large animal models in NF1 will undoubtedly be useful to (i) better understand the physiopathology of the disease (because the mouse model has limitations in reproducing human disease; the authors rightly point out, for example, that heterozygous NF1-/+ mice do not develop symptoms, unlike humans) and (ii) to study treatments.

In the abstract the authors evoke NF1, NF2 and schwannomatosis but they finally deal only with NF1. This should be clarified in the title and abstract.

In the description of genetically modified NF1 mouse models, it would be interesting to describe recent elegant models that have allowed to suggest new physiopathological hypothesis, e.g. they suggest that boundary cap cells could be involved in the development of nerve sheath tumors associated with NF1. Please cite:

- Radomska KJ, Coulpier F, Gresset A, Schmitt A, Debbiche A, Lemoine S, Wolkenstein P, Vallat JM, Charnay P, Topilko P. Cellular Origin, Tumor Progression, and Pathogenic Mechanisms of Cutaneous Neurofibromas Revealed by Mice with Nf1 Knockout in Boundary Cap Cells. Cancer Discov. 2019;9(1):130-147.

- Zhiguo Chen, Juan Mo, Jean-Philippe Brosseau, Tracey Shipman, Yong Wang, Chung-Ping Liao, Jonathan M. Cooper, Robert J. Allaway, Sara J.C. Gosline, Justin Guinney, Thomas J. Carroll and Lu Q. Le. Spatiotemporal Loss of NF1 in Schwann Cell Lineage Leads to Different Types of Cutaneous Neurofibroma Susceptible to Modification by the Hippo Pathway. Cancer Discov 2019;9(1):114-129

The MPNST model related to the inactivation of SUZ12 could also be cited, see:

De Raedt T, Beert E, Pasmant E, Luscan A, Brems H, Ortonne N, Helin K, Hornick JL, Mautner V, Kehrer-Sawatzki H, Clapp W, Bradner J, Vidaud M, Upadhyaya M, Legius E, Cichowski K. PRC2 loss amplifies Ras-driven transcription and confers sensitivity to BRD4-based therapies. Nature. 2014;514(7521):247-51.

Round 2

Reviewer 1 Report

The authors answered to all my concerns. I think thath the manuscript is now suitable for publication